# Molecular Profile and Prognostic Value of *BAP1* Mutations in Intrahepatic Cholangiocarcinoma: A Genomic Database Analysis

**DOI:** 10.3390/jpm12081247

**Published:** 2022-07-29

**Authors:** Alessandro Rizzo, Riccardo Carloni, Angela Dalia Ricci, Alessandro Di Federico, Deniz Can Guven, Suayib Yalcin, Giovanni Brandi

**Affiliations:** 1Struttura Semplice Dipartimentale di Oncologia Medica per la Presa in Carico Globale del Paziente Oncologico “Don Tonino Bello”, I.R.C.C.S. Istituto Tumori “Giovanni Paolo II”, Viale Orazio Flacco 65, 70124 Bari, Italy; 2Department of Specialized, Experimental and Diagnostic Medicine, University of Bologna, Via Giuseppe Massarenti, 9, 40138 Bologna, Italy; riccardo.carloni2@studio.unibo.it (R.C.); alessandrodifederico1@gmail.com (A.D.F.); brandi.giovanni20@gmail.com (G.B.); 3Division of Medical Oncology, IRCCS Azienda Ospedaliero-Universitaria di Bologna, Via Albertoni, 15, 40138 Bologna, Italy; 4Medical Oncology Unit, National Institute of Gastroenterology, “Saverio de Bellis” Research Hospital, 70013 Castellana Grotte, Italy; dalia.ricci@gmail.com; 5Department of Medical Oncology, Hacettepe University Cancer Institute, Ankara 06230, Turkey; denizcguven@hotmail.com (D.C.G.); suayibyalcin@gmail.com (S.Y.)

**Keywords:** intrahepatic cholangiocarcinoma, biliary tract cancer, *BAP1*, *FGFR2*, genomic, cholangiocarcinoma

## Abstract

Background. Recent years have witnessed the advent of molecular profiling for intrahepatic cholangiocarcinoma (iCCA), and new techniques have led to the identification of several molecular alterations. Precision oncology approaches have been widely evaluated and are currently under assessment, as shown by the recent development of a wide range of agents targeting Fibroblast Growth Factor Receptor (FGFR) 2, Isocitrate Dehydrogenase 1 (IDH-1), and BRAF. However, several knowledge gaps persist in the understanding of the genomic landscape of this hepatobiliary malignancy. Methods. In the current study, we aimed to comprehensively analyze clinicopathological features of *BAP1*-mutated iCCA patients in public datasets to increase the current knowledge on the molecular and biological profile of iCCA. Results. The current database study, including 772 iCCAs, identified *BAP1* mutations in 120 cases (15.7%). According to our analysis, no differences in terms of overall survival and relapse-free survival were observed between *BAP1*-mutated and *BAP1* wild-type patients receiving radical surgery. In addition, IDH1, PBRM1, and ARID1A mutations were the most commonly co-altered genes in *BAP1*-mutated iCCAs. Conclusions. The genomic characterization of iCCA is destined to become increasingly important, and more efforts aimed to implement iCCA genomics analysis are warranted.

## 1. Introduction

Biliary tract cancers (BTCs) include a heterogeneous group of aggressive and rare hepatobiliary tumors accounting for approximately 10–15% of primary liver cancers and 3% of all gastrointestinal malignancies [1]. BTCs encompass intrahepatic cholangiocarcinoma (iCCA), extrahepatic cholangiocarcinoma (eCCA), and gallbladder carcinoma (GBC). In recent years, the incidence of BTC has progressively increased worldwide, and a proportion of cases ranging from 60% to 70% is diagnosed with advanced stage—locally advanced or metastatic disease [2,3]. Most BTCs are diagnosed at a metastatic stage, and more than a decade after the publication of the ABC-02 trial establishing gemcitabine plus cisplatin as the front-line standard for metastatic BTC, the prognosis of this patient population remains poor [4,5]. Surgical resection represents the only potentially curative treatment option for BTC, but even following radical surgery with curative intent, 5-year overall survival (OS) is only 20–35% [6]. Recent years have witnessed the advent of molecular profiling in this setting, and new technologies have led to the identification of several molecular alterations in BTC [7,8,9,10,11,12]. Thus, precision oncology approaches have been widely evaluated and are currently under assessment, as shown by the recent development of a wide range of agents targeting Fibroblast Growth Factor Receptor (FGFR) 2, Isocitrate Dehydrogenase 1 (IDH-1), BRAF, and Human Epidermal Growth Factor Receptor 2 (HER2) [13,14,15,16,17,18,19]. However, most of the molecularly targeted therapies, including antiangiogenic agents, have shown limited efficacy in BTC, and several questions regarding the effective role of these anticancer treatments as well as the prognostic biomarkers remain unanswered [20,21].

*BRCA1*-associated protein 1 (*BAP1*) is a nuclear deubiquitinating enzyme in the ubiquitin C-terminal hydrolase family; this enzyme is involved in chromatin remodeling, together with other genes, including *ARID1A* and *PBRM1* [22]. Germline *BAP1* mutations have been associated with a hereditary cancer syndrome (the BAP1 tumor predisposition syndrome), characterized by an increased incidence of basal cell carcinoma, renal cell carcinoma, uveal melanoma, mesothelioma, and iCCA [23]. Similar to what is observed in TP53 germline mutations in the Li-Fraumeni syndrome, *BAP1* germline mutations are inherited as autosomal-dominant mutations with high penetrance, since more than 80% of carriers are destined to develop at least one of the main *BAP1*-associated malignancies throughout their life. Of note, the genomic analysis of *BAP1*-related tumors has highlighted the loss of the remaining *BAP1* wild-type allele, something that suggests the possible role as “two-hits” tumor suppressor gene by *BAP1* [24].

The last decade has seen a growing interest in the genomic characterization of iCCA and the identification of *BAP1* mutations, since these data, integrated with clinicopathological information, may drive the proper selection of patients that could derive benefit from novel, emerging treatments, such as targeted therapies, as well as the better estimations of prognosis to aid in treatment escalation and de-escalation. However, the available data on the prognostic role of BAP1 in iCCA are unequivocal due to the heterogeneous patient cohorts and modest sample sizes in most available studies, necessitating the evaluation of BAP1 in larger iCCA cohorts. In the current study, we aimed to investigate clinicopathological features of *BAP1*-mutated iCCAs in public datasets, exploring the prognostic value of these genetic aberrations in patients receiving radical surgery.

## 2. Materials and Methods

### 2.1. Clinical and Mutational Database

Data regarding clinical outcomes, mutational profiles, and copy number alterations in patients affected by iCCA were downloaded from the cBioPortal for Cancer Genomics Database; iCCA data were available from three trials and a data set, for a total number of 772 profiled samples [25,26,27] (Figure 1). Available clinicopathological data included diagnosis age, overall survival, disease-free survival, and gender. Mutation data were referred to the OncoKB knowledge base for disease-specific levels of evidence. Data were obtained from open access databases, and thus, no informed consent or statements of approval were required.

### 2.2. Statistical Methods

The Fisher’s exact test was used to analyze categorical data, and the univariate odds ratios were reported for the association. Continuous variables were compared by two-tailed Student’s t-test. We estimated and compared the survival curves by using Kaplan–Meier estimates of overall survival and log-rank tests. Cox proportional hazards model was used to analyze the association between clinical outcomes (disease-free survival and overall survival) and molecular data obtained from the cBioPortal database. Statistical analyses were performed by using SPSS (IBM SPSS statistics 25.0, Armonk, NY, USA). All statistical tests were two-sided, and the *p*-value < 0.05 was considered statistically significant. The Oncoprinter tool was used to graphically visualize the results [28,29].

## 3. Results

A total of 772 iCCA samples were available from the three public datasets in the cBioPortal database, and among these, 120 (15.7%) presented *BAP1* mutations (Table 1). *BAP1*-mutated iCCAs included 25 (20.8%) male and 48 (40%) female patients; for 47 (39.2%) patients, gender data were not available. Median age at diagnosis was 58 years (interquartile range (IQR) = 46–82 years). Among 120 *BAP1*-mutated iCCAs, 124 mutations were identified; most of these aberrations were missense driver mutations (Figure 2). Among all iCCAs with *BAP1* mutations, *IDH1* (26 patients, 21.8%), *PBRM1* (*n* = 16, 13.4%) and *ARID1A* (*n* = 15, 12.6%) represented the most frequently co-altered genes (Table 2); the 19.8% (*n* = 22) of *BAP1*-mutated samples presented *FGFR2* gene fusions.

Survival analysis was performed on *BAP1*-mutated and *BAP1* wild type patients for whom overall survival (OS) and relapse-free survival (RFS) were available and previously treated with radical surgery. Regarding OS, survival data were available for 81 *BAP1*-mutated and 433 *BAP1* wild-type iCCA patients. According to this analysis, no statistically significant differences were observed between the two groups, with a median OS of 35.55 months (27.24–44.98) and 26.81 (23.92–32.26) in *BAP1*-mutated and *BAP1* wild type iCCAs, respectively (*p* = 0.805) (Figure 3). Similarly, no statistically significant differences in RFS were highlighted between the same two groups, with a median DFS of 24.25 months (15.47–38.21) and 16.39 (12.75–23.95) in *BAP1*-mutated and *BAP1* wild type iCCAs, respectively (*p* = 0.508) (Figure 4).

## 4. Discussion

CCA remains a group of aggressive malignancies with heterogeneous anatomical, biological and clinical characteristics; regardless of anatomical distinction and location of CCA, surgical resection or orthotopic liver transplant are considered the potentially curative options for localized disease. At the same time, due to the lack of early screening methods and their aggressive biology, most patients are diagnosed with unresectable disease, a setting where limited treatment options are available. Thus, the grim prognosis of these tumors urgently calls for some improvements, and the better understanding of molecular pathology of CCA has opened the doors for precision medicine and targeted therapies in this setting. Herein, we analyzed the integration of clinicopathological and molecular data of iCCAs deposited in the public molecular cBioPortal for Cancer Genomics dataset. The current database analysis of the mutational profile of 120 *BAP1*-mutated iCCAs highlighted no differences between *BAP1*-mutated and *BAP1* wild-type patients receiving radical surgery in terms of OS and RFS (Figure 3; Figure 4). In addition, *IDH1*, *PBRM1*, and *ARID1A* mutations were the most commonly co-altered genes in *BAP1*-mutated iCCAs (Table 1; Table 2). The genomic characterization of iCCA is destined to become increasingly important, and more efforts aimed at implementing iCCA genomics analysis are warranted since genomic data are modifying the therapeutic management of these malignancies. First, the identification of *FGFR* fusions has been associated with the onset of iCCA, and two FGFR-targeted agents, infigratinib and pemigatinib, have been approved by the United States Food and Drug Administration (FDA) following the results of phase II trials assessing these *FGFR* inhibitors in previously treated patients with *FGFR2* gene fusions or rearrangements [30]. Similarly, two other *FGFR* inhibitors, futibatinib and derazantinib, have reported encouraging activity in iCCA and have the potential to enter soon in clinical practice as well [31]. Comprehensive genomic profiling studies have revealed the role of *BRAF* alterations in iCCA tumorigenesis, leading to the development of *BRAF* and *MEK* inhibitors in this setting and HER2 targeting approaches in *ERBB2* (*HER2*) amplifications [32]. *IDH* mutations have been also identified in iCCA, and *IDH* inhibitors such as ivosidenib have reported interesting activity for *IDH1* mutant patients progressing on first-line chemotherapy [33].

*BAP1* is a tumor suppressor gene whose loss can drive carcinogenesis in several tissues, and it currently represents one of the most attractive cancer-related genes for several reasons, including its presence in several solid tumors (for example, malignant pleural mesothelioma, basal cell carcinoma, squamous cell carcinoma, uveal melanoma, renal cell carcinoma, etc.), as well as the number of pathways that are indirectly or directly modulated by this gene (e.g., immune cells development, ferroptosis, cell metabolism, etc.) [34]. *BAP1* mutations are considered a relatively common finding in iCCA, with previous reports suggesting a percentage ranging from 15% to 35%. However, the prognostic role of *BAP1* mutations on survival outcomes in iCCA is conflicting. In one of the earliest reports, the presence of BAP1 mutations was associated with significantly lower PFS and OS. However, the small sample size (*n* = 75), the inclusion of both iCCA and eCCA, and a limited number of cases with BAP1 mutations (*n* = 7) limited the generalizability of the results [35]. Later, Boerner and colleagues evaluated the genomic landscape of 412 patients with iCCA and observed similar OS in patients with or without *BAP1* mutations (HR: 1.10, 95% CI: 0.79–1.51, *p* = 0.69) [20]. Interestingly, Jusakul et al. evaluated the epigenomic landscape of CCA and observed *BAP1* enrichment in cluster 4, which is among the clusters with a better prognosis [36]. Although further clinical and translational evidence is needed, the current study encompassing over 700 cases underlined no negative prognostic impact for *BAP1* mutations. Similar to unequivocal results with *BAP1* mutations, the *BAP1* expression had a complicated association with prognosis in CCA, and available studies reported both prolonged and decreased RFS and OS with *BAP1* expression loss, as well as no effect of *BAP1* expression on survival [37,38,39]. These data further support the complex role of *BAP1* in CCA prognosis and point out a need for further research [40,41,42,43]. At the same time, it is worth noting that available data remain still too scarce to investigate the association between *BAP1* mutations and OS and RFS in patients receiving radical surgery, and this link has never been confirmed. Another point to discuss is the percentage of *BAP1* mutations reported in iCCA patients. In fact, the effective incidence of these alterations is still a matter of debate in iCCA, since conflicting results have been highlighted across different studies, probably due to heterogeneity in methodology (for example, primary or metastatic tumors as well as the use of different techniques or the inclusion of different BTC subtypes) [44,45,46,47,48,49]. Our analysis, the largest available in literature so far, showed that BAP1 mutations were reported in 15.7% of iCCAs. Interestingly, available epidemiological data are not adequate to confirm the link between BAP1 germline mutations and CCA development, and further clinical evidence is needed to explore this link. For example, a report published by Pilarski and colleagues [43] detected the development of an adenocarcinoma of unknown primary—which was defined as from CCA—in a patient with a BAP1 germline truncating mutation (c.1182C>G, p.Tyr394*).

This database analysis has some limitations to be acknowledged, including the paucity of data in terms of survival, systemic treatments, and metastatic sites. In addition, it was possible to analyze OS and DFS only in a proportion of *BAP1*-mutated and *BAP1* wild type iCCA patients receiving radical surgery, and all our results derive from analyzing previous reports on this topic. Lastly, it was not possible to explore the prognostic impact of co-mutated genes such as IDH1, PBRM1, and ARID1A in this patient population due to the lack of data. This is particularly important since understating the role of co-occurring mutations and their impact on clinical outcomes and therapeutic response to medical treatments remains a clinical unmet need in this setting. Despite the limitations affecting our study, this large-scale database analysis may support the design of appropriate translational studies aimed at improving available data on *BAP1* mutations in iCCA and to better define the role of these aberrations in this patient population.

## Figures and Tables

**Figure 1 jpm-12-01247-f001:**
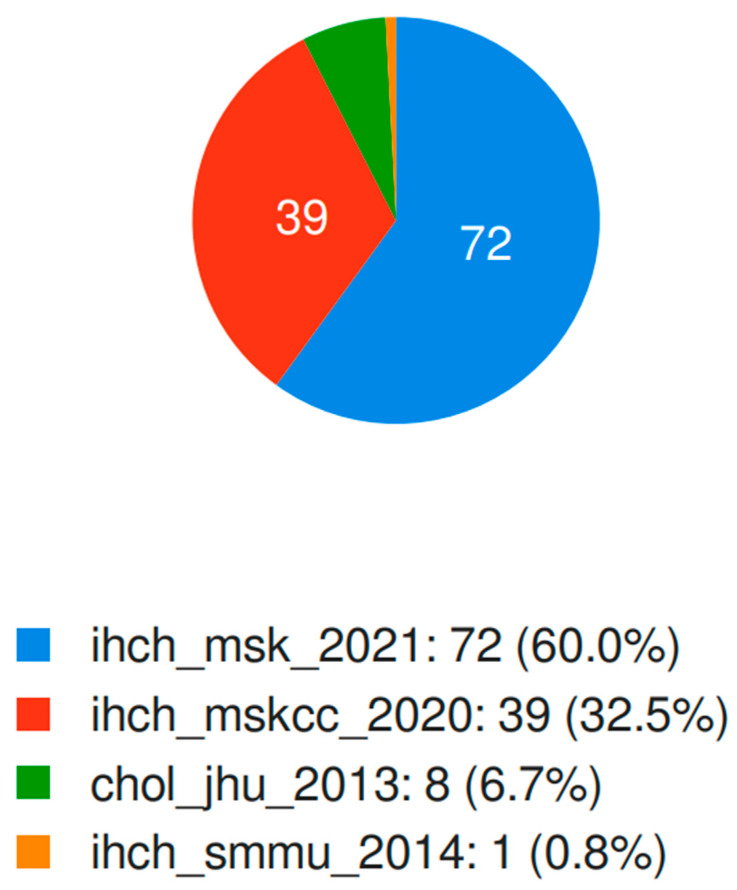
Schematic figure reporting the studies from which *BAP1*-mutated intrahepatic cholangiocarcinoma data were extracted.

**Figure 2 jpm-12-01247-f002:**
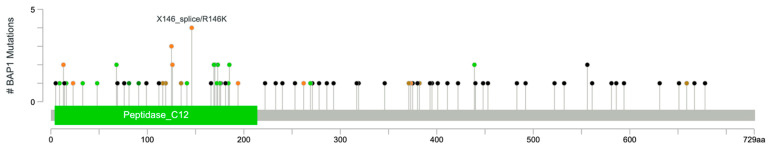
Lollipop plot showing the position of detected *BAP1* mutations in the gene sequence.

**Figure 3 jpm-12-01247-f003:**
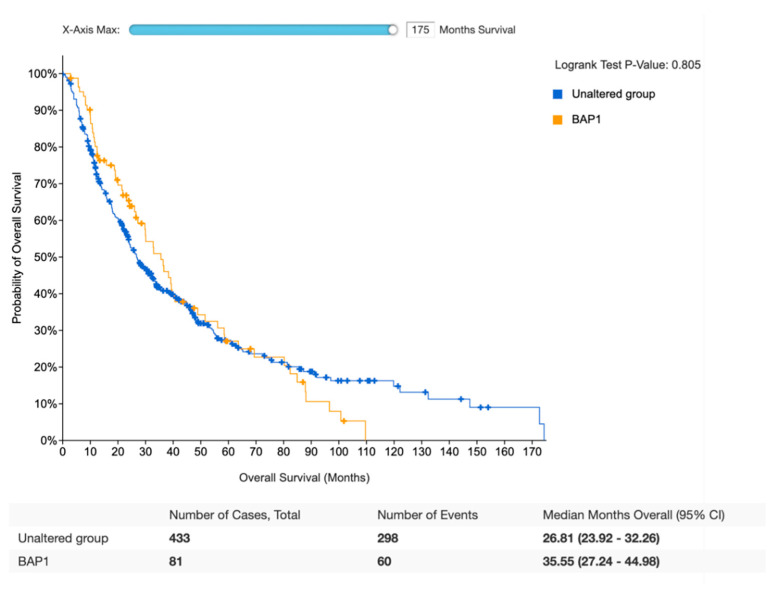
Kaplan–Meier analysis of overall survival of *BAP1*-mutant iCCA patients versus *BAP1* wild-type.

**Figure 4 jpm-12-01247-f004:**
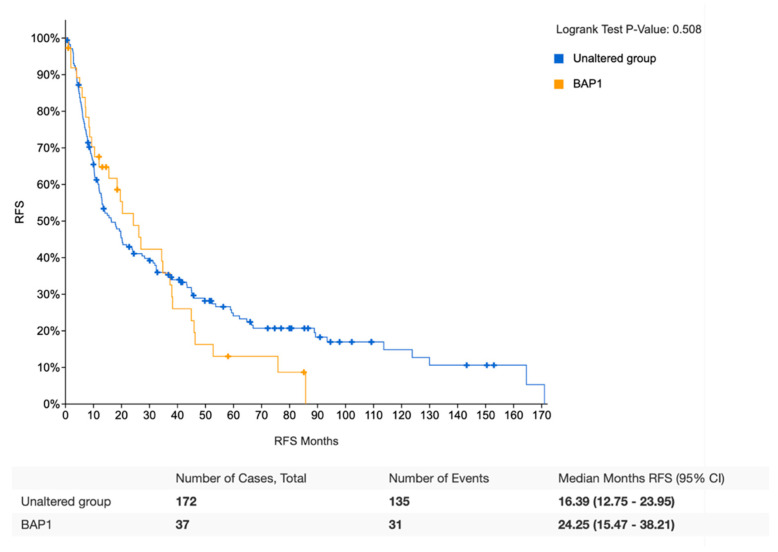
Kaplan–Meier analysis of relapse-free survival of *BAP1*-mutant iCCA patients versus *BAP1* wild-type.

**Table 1 jpm-12-01247-t001:** The most commonly mutated genes reported across 772 profiled samples.

Gene	Number of Samples/Patients	Frequency
*TP53*	152	19.9%
*IDH1*	139	18.2%
*ARID1A*	127	16.5%
*BAP1*	120	15.7%

**Table 2 jpm-12-01247-t002:** The most commonly co-altered genes in BAP1-mutated iCCAs.

Gene	Number of Samples/Patients	Frequency
*IDH1*	26	21.8%
*PBRM1*	16	13.4%
*ARID1A*	15	12.6%

## Data Availability

Not applicable.

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
