# Peer review of "Molecular Profile and Prognostic Value of BAP1 Mutations in Intrahepatic Cholangiocarcinoma: A Genomic Database Analysis"

_jpm, 2022, doi:10.3390/jpm12081247_

Round 1

Reviewer 1 Report

Dear Authors,

1) It is an interesting article. Since BAP1 mutations are renown in the field and you are trying to put together the data in the field for the iCCA, I think you should state more from the literature and have at least 15 more references for your article.

2) The type of article can become more like a meta-analysis. You should find other sources of data and integrate in your analysis where possible. Overall it is a good article.

Author Response

Dear Reviewer, thank you for your comments and for the time spent revising our paper.

  1. Thank you for this comment. We modified the manuscript according to your suggestions, as you could find in the revised paper. In particular, we added some references, as suggested. Our changes have been highlighted in green color. 
  2. Thank you for this comment. Unfortunately, it was not possible to "meta-analyze" the data extracted from each study due to heterogeneity. Thank you for your comprehension.

We hope the revised paper will better suit the journal. Thank you again for appreciating our work.

Reviewer 2 Report

The authors analyzed clinicopathological features of BAP1-mutated iCCA patients in public datasets including 772  iCCAs. They identifiedBAP1 mutations in 120 cases (15.7%). There is no difference of the overall survival and relapse-free survival between BAP1-mutated and BAP1 wild-type patients receiving radical surgery. In addition, they found that IDH1, PBRM1, and ARID1A mutations were the most commonly co-altered genes in BAP1-mutated iCCAs.

The weakness of the manuscript is that there is no original data from those authors. Also, those analyses are simple and do not provide an impressive novel discovery.

Author Response

Dear Reviewer, thank you for your comments and for the time spent revising our paper.

Our analysis was performed according to available studies and literature, and some data were not available. However, we extensively modified the paper, and we better discussed several crucial points in the Introduction and the Discussion section, including the limitations of the current study (blue and red color).

We hope the revised paper will better suit the journal. 

Reviewer 3 Report

In this work, the authors have analyzed mutational profiles of 772 Intrahepatic Cholangiocarcinoma (iCCA) patients, to investigate the prognostic relevance of BAP1 mutations (identified in 15% of cases) in this cancer. And found that there is no prognostic relevance of the presence of this mutation in this cohort of patients. They have mentioned of conflicting reports on such association, from other groups of investigators as well, in the Conclusion of the manuscript. In that light, the most obvious question arises as to why should this protein be focused on, at all?

Can the authors shed light on the timing of acquisition of BAP1 mutations in iCCA - is it detected as an early mutation, or rides along with other driver mutation in later stages of the disease?

Among the co-mutated genes, do any of those genes show prognostic relevance in iCCA patients? If so, can the samples be further stratified into groups based on mutual exclusivity of occurrence with BAP1?

Overall, I would request the authors to go for depth of the study, and try to investigate the samples from other possible angles.

Author Response

Dear Reviewer,

Thank you so much for the time spent revising our paper and for your valuable comments. In the revised manuscript, we modified several parts and paragraphs.

In particular, we better explained the rationale behind our research in the Introduction section and we expanded the Discussion section, also discussing some limitations of the current analysis (e.g., "In addition, it was possible to analyze OS and DFS only in a proportion of BAP1-mutated and BAP1 wild type iCCA patients receiving radical surgery, and all our results derive from analyzing previous reports on this topic. Lastly, it was not possible to explore the prognostic impact of co-mutated genes such as IDH1, PBRM1, and ARID1A in this patient population due to the lack of data. This is particularly important since understating the role of co-occurring mutations and their impact on clinical outcomes and therapeutic response to medical treatments remains a clinical unmet need in this setting. Despite the limitations affecting our study, this large-scale database analysis may support the design of appropriate translational studies aimed at improving available data on BAP1 mutations in iCCA and to better define the role of these aberrations in this patient population.").

Thus, it was not possible to explore the prognostic relevance of the co-mutated genes, and similarly, it was not possible to further stratify the groups. At the same time, the aim of our research was mainly to focus our attention on the prognostic value of BAP1 mutations, a controversial point in iCCA.

Moreover, we better discussed the features of BAP1 mutations, as follows: "Germline BAP1 mutations have been associated with a hereditary cancer syndrome (the BAP1 tumor predisposition syndrome), characterized by an increased incidence of basal cell carcinoma, renal cell carcinoma, uveal melanoma, mesothelioma, and iCCA [23]. Similar to what observed in TP53 germline mutations in the Li-Fraumeni syndrome, BAP1 germline mutations are inherited as autosomal-dominant mutations with high penetrance, since more than 80% of carriers are destined to develop at least one of the main BAP1-associated malignancies throughout their life. Of note, genomic analysis of BAP1-related tumors has highlighted the loss of the remaining BAP1 wild-type allele, something that suggests the possible role as “two-hits” tumor suppressor gene by BAP1 [24].". 

Thank you again for the time spent revising our paper. We hope the revised manuscript will better suit the journal.

Reviewer 4 Report

This work enlightens an important and urgent problem - the search for prognostic markers of intrahepatic cholangiocarcinoma. Intrahepatic cholangiocarcinoma (ICC) is one of the most aggressive tumors, accounting for about 10–15% of all primary malignant liver diseases. Radical resection remains the main treatment for patients with resectable lesions. Therefore, the search for prognostic molecular markers of this disease is extremely important. In the last decade, significant progress has been made in understanding the molecular genetic profile of intrahepatic cholangiocarcinoma (iCCA) and more than 21 risk markers have been identified. There are no fundamental comments to the article. The only thing is that the ethnicity of the patients is not entirely clear. Accounting for the relationship of ethnicity and frequencies of some SNPs will help to more accurately identify disease risk markers. 

Author Response

Dear Reviewer, 

Thank you for the time spent revising our paper and for appreciating our work.

Yes, data regarding ethnicity were not available and it was not possible to conduct this analysis. We are aware our results are preliminary and would deserve further exploration and validation. At the same time, we think that this large-scale database analysis may support the design of appropriate translational studies aimed at improving available data on BAP1 mutations in iCCA.

In the revised paper, we modified several parts and we better discussed the rationale of our study, as well as the limitations of the current paper.

Thank you again. We hope the revised manuscript will better suit the journal.

Round 2

Reviewer 2 Report

The revision greatly improved the quality of the manuscript. I would recommend accepting it for publication. 

Reviewer 3 Report

Please modify the Conclusion of the Abstract, as well as the end of the Introduction, to pinpoint on the utility of the study.